# Interactome of *Arabidopsis* ATG5 Suggests Functions beyond Autophagy

**DOI:** 10.3390/ijms241512300

**Published:** 2023-08-01

**Authors:** Pernilla H. Elander, Sanjana Holla, Igor Sabljić, Emilio Gutierrez-Beltran, Patrick Willems, Peter V. Bozhkov, Elena A. Minina

**Affiliations:** 1Department of Molecular Sciences, Uppsala BioCenter, Swedish University of Agricultural Sciences and Linnean Center for Plant Biology, 750-07 Uppsala, Sweden; pernilla.elander@hotmail.com (P.H.E.); sanjana.holla@slu.se (S.H.); igor.sabljic@slu.se (I.S.); peter.bozhkov@slu.se (P.V.B.); 2Instituto de Bioquımica Vegetal y Fotosıntesis, Universidad de Sevilla and Consejo Superior de Investigaciones Cientıficas, 41092 Sevilla, Spain; egutierrez@ibvf.csic.es; 3Departamento de Bioquimica Vegetal y Biologia Molecular, Facultad de Biologia, Universidad de Sevilla, 41012 Sevilla, Spain; 4Department of Plant Biotechnology and Bioinformatics, Ghent University, 9052 Ghent, Belgium; patrick.willems@psb.vib-ugent.be; 5Department of Biomolecular Medicine, Ghent University, 9000 Ghent, Belgium

**Keywords:** plant proteomics, plant ubiquitin-like conjugation system, autophagy-unrelated functions, nuclear ATG5, nuclear ATG12, posttranslational modifications, PP2A, HXK1, endomembrane trafficking, proteasome

## Abstract

Autophagy is a catabolic pathway capable of degrading cellular components ranging from individual molecules to organelles. Autophagy helps cells cope with stress by removing superfluous or hazardous material. In a previous work, we demonstrated that transcriptional upregulation of two autophagy-related genes, *ATG5* and *ATG7*, in *Arabidopsis thaliana* positively affected agronomically important traits: biomass, seed yield, tolerance to pathogens and oxidative stress. Although the occurrence of these traits correlated with enhanced autophagic activity, it is possible that autophagy-independent roles of ATG5 and ATG7 also contributed to the phenotypes. In this study, we employed affinity purification and LC-MS/MS to identify the interactome of wild-type ATG5 and its autophagy-inactive substitution mutant, ATG5^K128R^ Here we present the first interactome of plant ATG5, encompassing not only known autophagy regulators but also stress-response factors, components of the ubiquitin-proteasome system, proteins involved in endomembrane trafficking, and potential partners of the nuclear fraction of ATG5. Furthermore, we discovered post-translational modifications, such as phosphorylation and acetylation present on ATG5 complex components that are likely to play regulatory functions. These results strongly indicate that plant ATG5 complex proteins have roles beyond autophagy itself, opening avenues for further investigations on the complex roles of autophagy in plant growth and stress responses.

## 1. Introduction

Plants must endure a range of unfavorable environmental conditions to survive and propagate. Heat, drought, water logging, salinity, and pests are but a few examples of environmental stresses, which plants must cope with. By utilizing autophagy, plant cells can dispose of harmful cellular constituents and recycle the material for other purposes. During macroautophagy (hereafter autophagy), cellular constituents are engulfed by a de novo formed double-membrane vesicle called autophagosome and delivered to the lytic vacuole for degradation [1]. The autophagy pathway is regulated by approximately 40 autophagy-related (ATG) proteins, which include about 20 core ATGs generally grouped into five large complexes/systems essential for autophagic activity [2]. Central to the autophagic pathway is the lipidation of ATG8, which comprises a series of reactions enabled by ATG4, ATG7, ATG3, ATG10 and ATG5-ATG12/ATG16 complex culminating in the conjugation of ATG8 with phosphatidylethanolamine (PE). ATG8-PE is anchored on the membrane of the forming autophagosome facilitating the membrane elongation, cargo recognition and autophagosome trafficking [3,4]. 

In our previous study [5], we upregulated individually *ATG5* and *ATG7*, encoding for two core autophagy proteins required for the lipidation of ATG8. We discovered that upregulation of either of these two genes positively affected autophagic activity without impacting the transcription of the other genes participating in the ATG8-lipidation, indicating that ATG5 and ATG7 are rate-limiting components of the autophagic pathway. Interestingly, overexpression (OE) of these genes promoted plant growth, seed yield and longevity. Furthermore, OE plants developed more inflorescences and exhibited prolonged flowering time, thus producing more seeds. Furthermore, *ATG5* and *ATG7 OE* had a delayed onset of leaf senescence compared to wild-type (WT) plants and improved resistance to necrotrophic pathogens and oxidative stress [5].

Even though the observed phenotypic traits correlated with upregulated autophagic activity in the OE plants, we still could not decidedly claim that these traits were caused solely by upregulated autophagy. The question remained whether ATG5 and ATG7 might also act in other pathways.

Indeed, there are several examples of ATG proteins participating in autophagy-unrelated pathways. For instance, ATG8 orthologs in animal cells partake in intracellular trafficking and Golgi transport [6]. A plant ATG8 ortholog was shown to interact with ABNORMAL SHOOT3 at the late endosome to promote senescence by protein degradation, this interaction did not require ATG8 conjugation with PE [7]. Furthermore, a product of ATG5 proteolytic cleavage was shown to act as an apoptotic effector in animal cells [8].

*Arabidopsis* knockout (KO) mutants of core *ATG* genes show no discernible phenotypes at early developmental stages, when grown under standard conditions [5]. However, they undergo early senescence, display increased stress-susceptibility, and compromised immunity to necrotrophs compared to WT plants [5]. Interestingly, KO of different *ATG* genes display a range of phenotypes under autophagy-inducing stress conditions. For example, while *atg5* and *atg7* plants have similar phenotypes [5], *atg2* plants display more severe senescence and growth stagnation symptoms [9], whereas *atg9* plants have less severe symptoms [10]. Since autophagy is abrogated in all these mutants, the difference in the phenotypes might stem from autophagy-unrelated functions of the encoded ATG proteins, as we hypothesised for *ATG5* and *ATG7* OE plants.

To shed light on autophagy-unrelated functions of the ATG proteins, we began with identifying ATG5 interactome under standard growth conditions that do not boost autophagy. ATG5 is known to form complex with ATG12 and/or ATG16 [11,12]. In *Arabidopsis* cells most of ATG5 is covalently linked to ATG12 [5] via a C-terminal glycine residue of ATG12 bound to a side chain of a lysine residue of ATG5. An AlphaFold2-generated structure prediction for *Arabidopsis* ATG5 [13,14], revealed its structural similarities with the better characterized ortholog of ATG5 from yeast [11], enabling a prediction of Lys128 (K128) as the lysine residue binding to ATG12.

ATG16 in turn, binds non-covalently to ATG5 to form an ATG5-ATG12/ATG16 complex, which possesses E3-like ligase activity and is further referred to as “complete complex”. ATG16 recruits the complete complex to the phagophore, where its E3-like ligase activity is implemented to conjugate ATG8 to the resident PE [15]. Interestingly, ATG5 and ATG16 can interact in the absence of ATG12 to form a different complex [12], which in this study is named “minimal complex”.

Here, we identified proteins pulled-down with two types of bait: WT ATG5 capable of forming a complete ATG5-ATG12/ATG16 complex that is required for autophagy and the mutant ATG5^K128R^ that can form only the minimal ATG5-ATG16 complex (Figure 1A). We demonstrated only partial overlap between autophagy-competent ATG5 and autophagy-incompetent ATG5^K128R^ interactomes. This functional network unveils previously unknown interacting partners of plant ATG5 and suggests its autophagy-unrelated functions, thereby providing a new insight into diverse roles ATGs play in plant growth, development, and stress responses.

## 2. Results

### 2.1. Generation and Characterization of Arabidopsis ATG5-TAP Lines

To identify interactors of both complete (ATG5-ATG12/ATG16) and minimal (ATG5-ATG16) complexes (Figure 1A), we engineered genetic constructs encoding for Tandem Affinity Purification (TAP) tag [16] fused with the C-terminus of ATG5, ATG5^K128R^ mutant or GFP, placed under the control of 2x35S promoter or *ATG5* native promoter. We predicted that ATG5^K128R^ mutation impairs the conjugation of *Arabidopsis* ATG5 to ATG12 at the early steps of the autophagy pathway (Figure 1A).

The resulting constructs were introduced into *Arabidopsis* Columbia-0 (Col-0) WT or autophagy deficient *atg5-1* background. Two transgenic lines per construct showing expression detectable by immunoblot were selected for further studies. To confirm that ATG5^K128R^ mutation indeed impairs conjugation with ATG12, formation of the complete complex and thus abrogates autophagy, we firstly performed immunodetection of the TAP tag in the transgenic lines expressing WT form of ATG5 or its K128R mutant. Indeed, the band corresponding to ATG5-TAP-ATG12 conjugate was not detectable in the protein extracts from plants expressing ATG5^K128R^ (Figure 1B).

We further verified that the K128R mutant is not able to restore autophagy in *ATG5*-deficient cells. For this, we isolated protoplasts from *atg5-1* plants expressing GFP-ATG8a marker for autophagosomes. Protoplasts were transformed with plasmids encoding TagRFP- or TAP-tagged ATG5 or ATG5^K128R^ and treated with AZD8055 and Concanamycin A (ConA) to induce autophagy and block degradation of autophagic bodies in the vacuole [17]. Upon induction of autophagy, GFP-positive autophagic bodies accumulated in the vacuoles in the presence of ATG5, but not in the presence of ATG5^K128R^, corroborating the inability of this K128R mutant to form the functional complete complex required for autophagy (Figure 1C).

Finally, we assessed phenotypes of *atg5-1* plants complemented with ATG5-TAP or ATG5^K128R^-TAP driven by the native *ATG5* promoter. In contrast to WT and ATG5-TAP complemented line, the ATG5^K128R^-TAP complemented lines failed to rescue autophagy deficiency phenotype manifested in early senescence, smaller rosette leaves and reduced number of inflorescences (Figure 1D).

### 2.2. Interactomes of ATG5 and Its Complete and Minimal Complexes

#### 2.2.1. Affinity Purification

The aforementioned *Arabidopsis* lines expressing ATG5-TAP, ATG5^K128R^-TAP or GFP-TAP under control of 2x35S promoter in the WT (Col-0) background were used for the pull-down assay. Tandem affinity purification was performed as described earlier [18] on the protein extracts obtained from fully expanded leaves harvested at the bolting stage of plants grown under standard long day conditions. This developmental stage was previously shown to correlate with a low basal autophagy level [5] and was chosen to enrich for autophagy-independent interactors of ATG5. Pull-down for ATG5 and ATG5^K128R^ was performed in three biological replicates, each comprising a pool of more than 15 plants. The pull-down for the negative control (GFP-TAP) was performed in two biological replicates sampled in an identical manner. Purified proteins were analysed using liquid chromatography-tandem mass spectrometry (LC-MS/MS) and resulting data was analyses using MaxQuant and MSqRobSum [19].

#### 2.2.2. Detection of Post-Translational Modifications

ATG proteins are constitutively expressed and undergo various post-translational modifications (PTMs) allowing rapid adjustment of their activity during switches between normal and stress conditions. For instance, inhibitory phosphorylation of mammalian ATG5 and ATG12 by ATG1 kinase governs spatio-temporal control of ATG8 lipidation during phagophore expansion [20], while inhibitory acetylation of mammalian ATG5, ATG7, ATG8 and ATG12 by the p300 acetyltransferase [21] is reversed by deacetylase Sirt1 during autophagy induction [22].

The role of PTMs in plant autophagy regulation is scarce and even less is known about the role of PTMs in the autophagy-unrelated functions of the plant ATG proteins. Therefore, we searched our LC-MS/MS data for the PTMs reliably detectable by mass spectrometry and also relevant for autophagy regulation, i.e., phosphorylation and acetylation, and detected PTMs of ATG5 and ATG12 (Figure 2A, Appendix A). Interestingly, phosphorylation of Ser187 was identified only in ATG5, while acetylation of Lys183 was detectable on both ATG5 and ATG5^K128R^ indicating that the latter modification might also play a role in autophagy-independent function of ATG5. Additionally, phosphorylation of Ser7 was present at detectable levels on the ATG12A protein, indicating potential conservation of inhibitory phosphorylation of plant ATG12 under nutrient-rich conditions.

#### 2.2.3. Criteria for Identifying ATG5 Interactomes

In order to identify specific and common interactors of the minimal and the complete complexes, we selected proteins enriched in ATG5^K128R^ and/or in ATG5 samples compared to the GFP-control. Significant interactors of ATG5/ATG5^K128R^ were identified using MSqRobSum (see Methods), requiring a minimal fold change of 1.5 (compared to GFP control) and *p* value < 0.05. After filtering, we retained 85 interactors only present in the ATG5 pull-down, 82 only in the ATG5^K128R^ pull-down and 88 common interactors in both ATG5 and ATG5^K128R^ pull-downs (Figure 2B). From here on, we grouped potential interactors into three categories: proteins pulled-down with ATG5 only (complete complex-specific), with ATG5^K128R^ only (potential interactors of individual ATG5) and with both ATG5 and ATG5^K128R^ (minimal complex-specific) (Figure 2C,D).

In accordance with conditions used for the affinity purification experiment, the typical autophagy-related interactors of ATG5, including ATG3, ATG10 and ATG8 [12], were not detectable in our samples, confirming that the identified candidate interactors were likely enriched for autophagy-unrelated pathways. Expectedly, ATG12A and ATG12B were only detected in the interactome of the complete complex, whereas ATG16 was detected in both interactomes (Figure 2C,D).

#### 2.2.4. Putative Interactors Acting as Stress Sensors

Interestingly, we identified PP2A-A3, the scaffolding subunit of the Protein Phosphatase 2A (PP2A), as an interactor of the minimal complex, while a catalytic subunit of PP2A (PP2A-3) was pulled down with autophagy-incompetent ATG5^K128R^. In mammalian cells PP2A is known to be an important regulator of the above mentioned PTMs, namely it dephosphorylates ATG13 upon inhibition of the TORC1 kinase complex enabling formation of ATG1-ATG13 complex and induction of autophagy [23]. Furthermore, different subunit compositions of the heterotrimeric PP2A have been shown to play alternative roles in animal autophagy [24]. Although plant PP2A is known to be important for abiotic and biotic stress response [25], its role in plant autophagy is still unknown and our observations provide the first insight on its potential participation and indicate a possible implication of autophagy-unrelated crosstalk between ATG5 and PP2A.

In addition to PP2A, we identified hexokinase 1 (HXK1, Appendix A) as a potential interactor of the minimal complex. HXK1 was previously shown to suppress plant autophagy under glucose-rich conditions via an unknown mechanism [26]. Our finding suggests that HXK1 might not be acting via TORC1 as suggested previously but rather through the direct interaction with ATG5 and/or ATG16.

#### 2.2.5. Putative Interactors belonging to Endomembrane Trafficking System

Autophagy is an integral part of the endomembrane trafficking system [27,28] and so far, ATG8 has been the best characterized molecular link between plant autophagosomal structures and other components of the endomembrane trafficking system [29]. Remarkably, we discovered a large set of proteins playing a role in endomembrane trafficking that interact with ATG5 and its complete and minimal complexes: proteins involved in COPII-mediated endoplasmic reticulum (ER) to Golgi transport (SAR1B, SAR1C, SEC24C, TRAPPC3), phagophore formation (TRAPPC11, SH3P2), ER-PM contact sites (SYT1, TPLATE subunit TML), retromer complex subunits (SNX1 and SNX2a) and a CLASP protein involved in membrane loading of SNX1 (Figure 2C,D, Appendix A). Most interestingly, our results showed that homologous proteins encoded by multi-member gene families, with not yet fully resolved redundancy, show specificity towards either the WT or autophagy incompetent ATG5 bait. For example, SAR1C was detectable in the ATG5^K128R^ pull-down only, unlike SAR1B that was found in both. Similarly, SNX1 was detected only in ATG5 pull-down samples, while SNX2a only in ATG5^K128R^ (Figure 2C,D, Appendix A).

#### 2.2.6. Putative Interactors belonging to Ubiquitin-Proteasome System

Autophagy is tightly interlinked with another catabolic pathway, governed by the ubiquitin-proteasome system (UPS) [30,31]. Furthermore, mammalian ATG5 was shown to directly interact with UPS components to aid mitophagy [32]. Our affinity purification assay allowed detection of Ubiquitin Fusion Degradation 1 (UFD), a component of the CDC48 complex that was previously suggested to crosstalk with autophagy to help maintenance of chloroplastic proteins during oxidative stress [33] (Appendix A). Furthermore, surprisingly, we discovered selective interaction between 20S proteasome subunit alpha E2 and complete ATG5 complex, while subunits alpha D1 and beta G1 showed preference towards the minimal complex (Appendix A).

#### 2.2.7. Putative Interactors of Nuclear-Localized ATG5 and ATG12

Interestingly, we also identified two components of nuclear pore complex as potential interactors of individual ATG5: Nucleoporin 155 (NUP155) and SEC13B (Figure 2D, Appendix A) [34]. The mammalian ortholog of ATG5 has been shown to translocate to the nucleus under stress conditions and play a role in arresting cell division [35]. Furthermore, deacetylation-regulated nuclear export of a mammalian ATG8 ortholog was suggested to be implicated in autophagy [36]. In addition, plant ATG8s were previously observed in both cytoplasm and nuclei [5,17,37]. Therefore, we decided to investigate if the plant ATG5 complex components might also be localizing to the nuclei. For this we compared localization of ATG5, ATG8, ATG12 (A and B isoforms), and ATG16 fluorescent fusions transiently expressed in *Nicotiana benthamiana* epidermal leaf cells (Figure 3). To our surprise, ATG5 and both ATG12 isoforms, but not ATG16, could indeed localize to the nuclei under normal conditions, similarly to the previously observed localization of ATG8 (Figure 3A).

To further elucidate whether ATG5 and ATG12 localize to the nucleus in a form of the stable ATG5-ATG12 conjugate, we checked localization of the ATG5^K128R^ defective in conjugation to ATG12 and of the of ATG12A-GFP, in which GFP impedes conjugation with ATG5. We observed that ATG5^K128R^ and ATG12A-GFP still could be found in the nuclei (Figure 3B), suggesting that conjugation is dispensable for nuclear localization of these proteins. These experiments were performed using GFP, YFP and TagRFP fusions of the proteins to ensure that observed localization was not an artefact caused by the fluorescent tag (only representative data for one of the fusion types is shown for each protein of interest). Finally, we used Western Blot analysis to exclude the possibility that nuclear signal observed by confocal microscopy resulted from passive diffusion of the fluorescent tag cleaved off the expressed fusion proteins. Detection of GFP and TagRFP in the protein extracts from leaves transiently expressing fluorescent fusions of ATG5, ATG12 and ATG16 confirmed the presence of intact fluorescent fusions (Figure 3C). In sum, these results demonstrated that at least three components of the autophagy ubiquitin-like conjugation system (ATG5, ATG12 and ATG8) localize to the nuclei. Furthermore, our affinity purification assay indicates that ATG5 might shuttle between nuclei and cytoplasm through interaction with the nuclear pore complex components NUP155 and SEC13b.

The molecular mechanisms underpinning typical autophagy-deficient plant phenotypes such as premature senescence and early onset and cessation of flowering have been only partially explained [9,38,39]. Our pull-down assay indicated interaction between ATG5 and nuclear proteins involved in stress response, flowering and photomorphogenesis (Appendix A). Those include HAM1, a catalytic subunit of NuA4 acetyltransferase complex with a role in chromatin remodelling during environmental response of *Arabidopsis* [40], lectin EULS3 involved in osmotic stress response and ABA signalling [41] and Nuclear Factor YC protein 3 (NF-YC3) playing a role in photomorphogenesis, drought response, flowering and ABA signalling [42,43], COP9 signalosome subunit 1 involved in photomorphogenesis [44] and CULLIN-ASSOCIATED AND NEDDYLATION DISSOCIATED 1 (CAND1), a known mediator of auxin signaling and flowering [45]. 

## 3. Discussion

### 3.1. Evolutionary Context of Autophagy

Compartmentalization of biosynthesis and catabolism in eukaryotic cells lead to development of an intricate endomembrane trafficking system comprising a set of membrane-bound organelles, each having a specialized function in the multistep delivery of cellular materials to their respective destinations. For example, proteins synthesized on the ER can be delivered to the lytic compartment or the plasma membrane (PM) via stepwise transport through the Golgi apparatus and the endosomal system [46]. Similarly, degradation of the cellular content might also rely on directed trafficking towards a specialized compartment [46,47].

Autophagy and the ubiquitin-proteasome system (UPS) are the two major catabolic pathways in eukaryotic cells that counterbalance biosynthesis. UPS, which relies solely on protein-based molecular machinery for substrate recognition, labelling and degradation [48], is likely a more ancient catabolic process compared to the autophagic pathway. Unlike UPS, autophagy [48] is an integral part of the endomembrane trafficking system and requires formation of a membrane compartment (autophagosomes) to sequester cargo and deliver it for degradation to the lytic compartment, and thus would not be feasible in prokaryotic organisms lacking endomembrane compartments. Indeed, a prototype of the eukaryotic UPS-like system was discovered in *Archaea* [49], while autophagy is known to be typical for eukaryotic organisms [43].

A significant increase in cell structure complexity of eukaryotes created a necessity for a catabolic pathway able to degrade such cargoes as large as complete organelles. Autophagy was possibly established as such endomembrane trafficking-based high-throughput catabolic mechanism in the last eukaryotic common ancestor (LECA) [43], which already possessed a functioning UPS. Subsequently, although UPS and autophagy are often referred to as separate catabolic pathways, they might actually comprise two branches of a single proteolytic network in which autophagy was established after UPS. Co-evolution of both pathways sculpted intricate mechanisms of mutual regulation, implementing common molecular tools such as ubiquitin-like conjugation systems and overlapping degrons [50].

Since ATG proteins have also evolved as an integral part of the endomembrane trafficking system, it is not unlikely they might participate in the routes of the endomembrane trafficking that are not directly linked to autophagy, in its classical depiction. One such example is the cytoplasm-to-vacuole targeting(cvt) pathway in yeast. This is the only known biosynthetic pathway that utilizes autophagic machinery, where core ATG proteins, with a minor change in complex formation deliver enzymes to the vacuole [51]. Furthermore, the endomembrane systems of plant, animal and fungi have significant structural and functional differences, e.g., the lytic compartments are represented either by vacuoles or by lysosomes, the Golgi apparatus is either a static ribbon like structure or mobile stack or even individual cisternae, and the formation and function of endosomes varies significantly [52,53,54,55]. The evolution of autophagy as the integral part of such endomembrane systems likely diverged to develop kingdom-specific features. There are two pending questions. First, how did the evolution of the endomembrane system impact autophagy-unrelated functions of ATG proteins while preserving their role in autophagy and cross-talk with UPS? Second, did ATG proteins carry over some autophagy-unrelated functions from their distant homologs found in prokaryotes [43]?

### 3.2. Closely Related Components of Endomembrane Trafficking Pathways Show Selectivity towards Different ATG5 Baits

In this study we initiated a discovery proteomics approach to expand our understanding of plant ATG5 in autophagic pathway and beyond it. By comparing proteins pulled-down using WT or autophagy-incompetent K128R mutant of ATG5 as a bait we identified potential interactors of complete and minimal ATG5 complexes, and of individual ATG5.

Remarkably, we identified examples of close homologs presumably involved in the same pathway interacting with ATG5 but showing different preferences towards autophagy-competent or autophagy-incompetent forms of the bait. These results can be further developed into a new tool to elucidate specific functions of these homologs and bring important insights on the roles of ATG5 in and beyond autophagy.

For example, Secretion-Associated-Ras-related GTPase (SAR1) plays a crucial role in the initiation of COPII vesicles formation and thus enables ER to Golgi trafficking [56]. *Arabidopsis* genome encodes five paralogs of SAR1A-E with suggested tissue specific expression and at least partial functional diversity [57,58,59]. Only SAR1B and SAR1D were shown to play a role in autophagosome maturation [60] and autophagosome biogenesis [61], respectively. Our pull-down assay identified SAR1B as an interactor of the minimal ATG5 complex and SAR1C as an interactor of specifically ATG5^K128R^. The latter interaction was most likely to be an example of autophagy-independent role of ATG5, as SAR1C was previously shown to not co-localize with autophagic structures under autophagy-inducing conditions [61].

Another example of such specificity towards the two ATG5 baits are sorting nexins (SNXs), subunits of a conserved among eukaryotes retromer complex, which is essential for retrograde trafficking and autophagy [62,63]. Elucidating the interactome of plant SNXs is especially interesting, as, despite being conserved, they seem to be dispensable for the functionality of the plant retromer complex [64] and might have evolved plant-specific functions instead [65]. In this study, we observed SNX1 being pulled-down only with the WT form of ATG5 (complete complex) and SNX2A only with the ATG5^K128R^ (individual ATG5). Additionally, we also identified CLASP protein required for SNX1 membrane association [66] as a potential interactor of the ATG5^K128R^. Future studies will help to elucidate the mechanism behind these intriguing specific preferences of SNX1 and SNX2A towards either WT or K128R mutant of ATG5, respectively, and potentially bring better understanding of the individual roles of SNXs.

Additionally, in this study, out of three existing *Arabidopsis* paralogs of SEC24 [57] which is a COPII coat complex component regulating vesicle formation during ER to Golgi trafficking, we identified only SEC24C as an interactor of complete ATG5 complex. Interestingly, the closest mammalian ortholog of SEC24C was shown to play a role in selective autophagy of ER (ER-phagy) [67]. Future studies involving ER-phagy-inducing conditions will help to assess the potential role of SEC24C in plant ER-phagy and the role of ATG5 in it.

### 3.3. Cross-Talk of ATG5 and Clathrin-Mediated Trafficking Is Conserved

Lastly, we uncovered a set of ATG5 interactors that are associated with ER-PMCS (endoplasmic reticulum- plasma membrane contact sites) and clathrin-mediated vesicular trafficking (TPLATE COMPLEX MUNISCIN-LIKE (TML), SH3 DOMAIN-CONTAINING PROTEIN2 (SH3P2), and SYNAPTOTAGMIN 1 (SYT). All three proteins are known to partake in autophagy [68,69,70]. However, our study provides novel evidence of their interaction with the ATG5 complexes. Interestingly, a recent publication on the ATG5 interactome in mice [12] also presented several interactors of ATG5 related to clathrin-mediated trafficking, linking this pathway to animal autophagy.

## 4. Methods

### 4.1. Plasmids Construction

For generation of 2x35S:ATG5-TAP constructs, the ATG5 gene AT5G17290.1 without UTRs was amplified using PE7 and PE8 primers (Appendix A) and cloned into pDONR/Zeo vector using Gateway cloning system (Invitrogen/Thermo Fisher Scientific, Waltham, MA USA). The point mutation was introduced into this entry clone using primers PE10 and PE11 (Appendix A) and QuikChange II Site-Directed Mutagenesis Kit (200523, Agilent Technologies, Inc., Santa Clara, CA, USA), following the kit’s instructions. The obtained entry clones with WT and K128R gene versions of ATG5 were then recombined into pCTAP vector (pYL436) under the control of 2x35S promoter [16] using Gateway cloning system (Invitrogen).

For generation of ATG5pr::ATG5-TAP constructs, the ATG5 gene AT5G17290.1 together with the promoter region was amplified using primers PE3 and PE4 (Appendix A). The TAP tag was amplified from the pCTAP vector [16] using primers PE5 and PE9 (Appendix A). The ATG5 gene and TAP tag were then fused using overlay PCR with primers PE3 and PE9 (Appendix A). The obtained amplicon was cloned into pDONR/Zeo vector using Gateway cloning system (Invitrogen). The point mutation was introduced into this entry clone using primers PE10 and PE11 (Appendix A) and QuikChange II Site-Directed Mutagenesis Kit (Agilent, 200523), according to the standard protocol. The obtained entry clones with WT and K128R gene versions of ATG5 genes including promoter region were then recombined into pGWB401 vector (Adgene, Plasmid #74795) using Gateway cloning system (Invitrogen).

Constructs for expressing fluorescent fusions of ATG5, ATG8, ATG16, ATG12A and ATG12B were produced using Gateway cloning system (Invitrogen). The corresponding genes or Coding DNA sequences were lifted from genomic DNA or total cDNA of *Arabidopsis* using primers provided in the Appendix A (PE8, SH139-145, AM 475, 476). Amplicons were recombined into pDONR/Zeo vecor to produce entry clones (see Appendix A: SH 210-213, SH 231, 232 and AM 655). The entry clones were later recombined into pGWB-series destination vectors carrying 2x35S promoter and GFP, YFP or TagRFP tags (Adgene) (see Appendix A, destination clones for expressing fluorescent fusions in plants).

The destination clones were used to transform *Agrobacterium tumefaciens* strain GV3101.

### 4.2. Protoplast and Plant Transformation

#### 4.2.1. Transient Expression of *Arabidopsis* Protoplasts

Protoplasts were isolated from leaves of four-weeks-old plants expressing GFP-ATG8 in *atg5-1* using the “Tape-*Arabidopsis* Sandwich” method described in [71]. The isolated protoplasts were transformed using 10–20 µg of each plasmid (Appendix A) using the method described in [72]. The transfected protoplasts were incubated in 24-well glass bottom plates (VWR CORN4441) for 16 h in light. The protoplasts were further treated with 5 µM AZD-8055 (364424, Santa-Cruz Biotech, Dallas TX, USA) and 0.5 µM Concanamycin A (202111A, Santa-Cruz Biotech) for 24 h, where applicable. The transformed protoplasts were imaged using CLSM800 (Carl Zeiss AG, Oberkochen BW, Germany), objective C-Apochromat 40×/1.2 W, excitation light 488 nm and 561nm and emission ranges of (515–560 nm) and (570–650 nm) for GFP and TagRFP, respectively. Images were analyzed using ZEN blue software (Carl Zeiss).

#### 4.2.2. Transient Expression in Nicotiana Benthamiana

*N. benthamiana* plants were grown in 8 cm^3^ pots filled with soil S-Jord (Hasselfors) under controlled growth conditions of 16 h light 8 h dark cycles, 70% relative humidity, light intensity of 150 μE m^–2^ s^–1^, and day and night temperature of 22 °C and 20 °C, respectively. Transformed *A. tumefaciens* strain GV3101 (Rifampicin, 100 μg/mL) carrying the constructs of interest were grown in 5 mL of Luria-Broth high salt medium (L1704, Duchefa Biochemi, Haarlem, The Netherlands), supplemented with appropriate antibiotics (see Appendix A, destination clones for expressing fluorescent fusions in plants). Liquid bacterial cultures were shaken at 200 rpm, 28 °C overnight, and then sedimented at 4000 G for five minutes. The resulting pellets were resuspended in MQ water with 150 μM Acetosyringone to the final OD_600_ = 0.15 and infiltrated in the abaxial side of leaves of five-week-old *N. benthamiana* plants. The leaves were imaged on the third day post-infiltration using CLSM800 (Carl Zeiss), objective C-Apochromat 40×/1.2 W, excitation light 488 nm and 561 nm and emission ranges of (515–560 nm) and (570–650 nm) for GFP and RFP, respectively. Images were analyzed using ZEN black software (Carl Zeiss).

#### 4.2.3. *Arabidopsis* Thaliana Growth and Transformation

*Arabidopsis* plants were grown in 8 cm^3^ pots filled with soil S-Jord (Hasselfors), under long day conditions: 150 µE m^−2^ s^−1^ light for 16 h, 8 h dark, 22 °C, 70% humidity.

*Arabidopsis* Col-0 wild-type plants and the previously described *atg5-1* mutant [73] were transformed using the standard “floral dip”-method [74]. *A. tumefaciens* strain GV3101 carrying the pYL436 and pGWB401 constructs (Appendix A) was used for transformation. Transgenic plants were selected on Murashige and Skoog (MS) medium containing 40 µg mL^−1^ gentamycin and 50 µg mL^−1^ kanamycin, respectively. The plants were genotyped to confirm the presence of the transgenes using PE7 and PE8 primers (Appendix A). Expression of the transgenes was confirmed using Western blot analyses (as described in the Section 4.3).

### 4.3. Immunoblotting

Plant material was powdered in liquid nitrogen, mixed with 2 vol. of hot 2× Laemmli buffer, and boiled for 10 min. Debris was pelleted for 5 min at 17,000 *G*. Proteins were separated on Mini-PROTEAN TGX stain-free Bio-Rad gels, 7.5% (Bio-Rad, Hercules, CA, USA) and transferred onto PVDF membranes. Membranes were blotted anti-actin 1:2000 (AS13 2640, Agrisera AB Vännäs, SWEDEN.), anti-myc 1:1000 (11667203001, Roche, Basel, Switzerland). Reactions were developed using Amersham ECL Prime kit (RPN2232, Cytiva, Marlborough, MA, USA) and detected using BioRad ChemiDoc.

### 4.4. Tandem Affinity Purification

Plants for affinity purification assay were grown in 8 cm^3^ pots filled with soil S-Jord (Hasselfors) under long day conditions: 150 µE m^−2^ s^−1^ light for 16 h, 8 h dark, 22 °C, 70% humidity.

Plant rosette leaves were harvested at bolting stage and flash frozen in liquid nitrogen. Sample weight was >15 g of frozen tissue. Subsequent procedure was performed as previously described [18] using the following consumables: Protease inhibitor cocktail (P9599-5ML, Sigma-Aldrich, Saint Louis, MO, USA), PreScission protease (GE27-0843-01, Sigma-Aldrich), Ig-G sepharose beads (GE17-0969-01, Sigma-Aldrich),), Ni-sepharose beads (GE17-5318-01, Sigma-Aldrich).

Mass spectrometry data was acquired using a data-dependent acquisition procedure with a cyclic series of a full scan from 350–1500 with resolution of 120,000 control (AGC) target 1E6, maximum injection time 100 ms. The top S (3 s) and dynamic exclusion of 30 s were used for selection of Parent ions for MSMS in the HCD cell with, relative collision energy 30% and scanned in the orbitrap with resolution of 30,000.

### 4.5. Mass Spectrometry

#### 4.5.1. Liquid Chromatography and Mass Spectrometry Assay

LC-MS/MS was performed at the Mass Spectrometry Facility of Rutgers Center for Advanced Biotechnology and Medicine, US. Samples were loaded on to a fused silica trap column Acclaim PepMap100, 75 µm × 2 cm (Thermo Fisher Scientific, Waltham, MA USA).). After washing for 5 min at 5 μL/min with 0.1% TFA, the trap column was brought in-line with an analytical column (NanoeaseMZ peptideBEH C18, 130A, 1.7 µm, 75 mm × 250 mm, Waters, Milford, MA, USA) for LC-MS/MS. Peptides were fractionated at 300 nL/min using a segmented linear gradient 4–15% B in 30 min (where A: 0.2% formic acid, and B: 0.16% formic acid, 80% acetonitrile), 15–25% B in 40 min, 25–50% B in 44 min, and 50–90% B in 11 min. The column was re-equilibrated with 4% B for 5 min prior to the next run.

#### 4.5.2. LC-MS/MS Data Analysis

Raw data files from the LC-MS/MS analysis were processed using MaxQuant (ver. 1.6.17.0) [19]. Default settings were used except enabling the protein label-free quantification (LFQ) and matching-between-runs options on default settings. Proteomics data was searched against the 26,755 representative Araport11 proteins [75] supplemented with the contaminant proteins list included within MaxQuant.

For the search including PTMs, phosphorylation (STY), acetylation of the side chain of Lys residues and protein N-terminal acetylation were specified in MaxQuant as variable modifications, while cysteine carbamidomethylation was set as fixed. Additionally, multiple events of methionine oxidation were detected, but not taken into consideration for this study, as it was not possible to discern whether they occurred in vivo or during samples processing [76].

#### 4.5.3. Statistical Analysis of Interactors—MSqRobSum

The obtained MaxQuant peptide and protein result tables were used for statistical analysis in R (ver. 4.02) by MSqRobSum (ver. 0.0.0.9000) [77]. Proteins only identified by site and contaminant/decoy proteins were filtered. After default preprocessing, protein intensity summaries were estimated by robust regression, fitting log_2_ intensities in function of condition (WT ATG5, PM ATG5, and GFP control). Afterwards, user-defined contrasts of interest were tested to identify wild-type ATG5 (WT ATG5/GFP control) and mutated ATG (PM ATG5/GFP control) interactors. Interactors were filtered as having a fold change ≥ 1.5 and *p* value ≤ 0.05. In addition, proteins not quantified in GFP control conditions but identified in at least 2 out of 3 replicates of a ATG5 pulldown were manually curated as putative ATG5 interactors.

#### 4.5.4. STRING Analysis

STRING analysis was made using https://string-db.org/ (accessed on 22 November 2022) website [78] with following parameters: full STRING network, active interaction sources: all possible; minimum required interaction score: medium confidence (0.400). Such settings were selected to include also possible connections that are not yet properly verified, which we found most suitable in the context of our search for unconventional roles of the ATG proteins. Suggested by STRING analysis connections were manually verified in the existing literature. When clustering was used it was with MCL clustering with an inflation parameter of 3.

## 5. Conclusions

This study provides novel evidence of PTMs occurring on core ATG proteins, i.e., ATG5 and ATG12 under non-inducing conditions. This discovery opens up exciting opportunities for further investigation and deeper understanding of the regulatory mechanisms governing ATG proteins in plants. Moreover, the comparative affinity purification assay revealed a compelling list of interactors for complete and minimal ATG5 complexes, indicating the possibility of crosstalk between autophagy-related proteins and multiple components of the endomembrane trafficking system and UPS. These findings strongly suggest the existence of shared and coordinated mechanisms that integrate these crucial cellular processes. Significantly, the identification of unique interactors for both ATG5 and ATG5^K128R^ suggests that ATG5 may possess roles and functions that extend beyond its classical involvement in autophagy. This finding underscores the complexity and versatility of ATG proteins and implies their potential contributions to diverse cellular pathways. To deepen our functional understanding of the complete and minimal ATG5 complexes, future studies should focus on verifying the identified interactors and conducting functional analyses of the candidate protein complexes. Additionally, we present novel evidence of nuclear localization for individual ATG5 and ATG12, that does not depend on their conjugation. Furthermore, we identified a set of nuclear-localized proteins, potential interactors of the nuclear fraction of ATG5 and ATG12. This discovery significantly expands the range of suggested functions associated with the components of the ATG5 complex. We hope that this study will pave the way for further explorations and place autophagic pathway into a broader context.

## Figures and Tables

**Figure 1 ijms-24-12300-f001:**
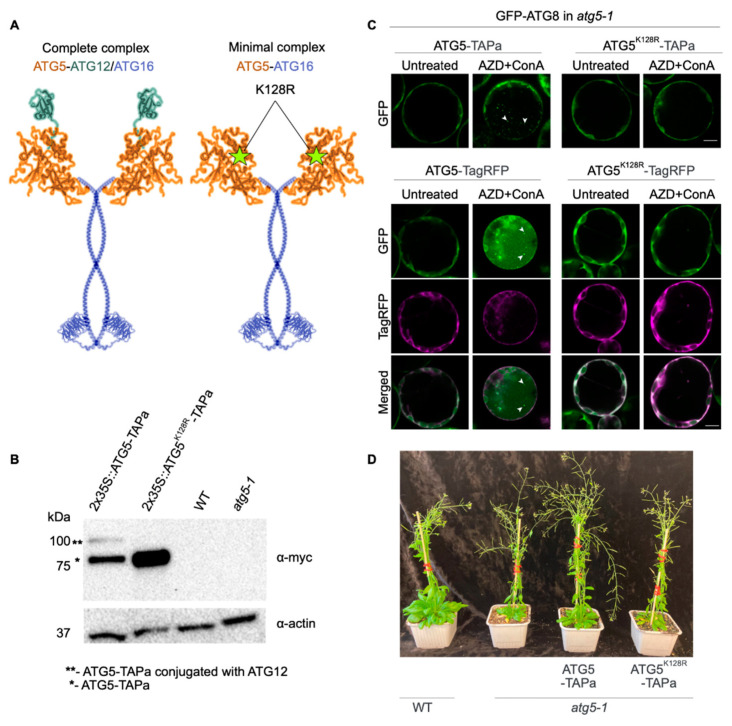
**ATG5 K128R mutation abrogates conjugation to ATG12 and formation of the complete complex.** (**A**). A schematic representation of the two complexes used to identify putative autophagy-unrelated interactors of ATG5: the complete complex encompassing ATG5-ATG12/ATG16 and a minimal complex comprising ATG5-ATG16. The K128R point mutation was predicted to impede covalent conjugation of ATG5 to ATG12 leading to the formation of only minimal ATG5-ATG16 complex. The complete complex is essential for autophagy, while the minimal complex is more likely to have autophagy-unrelated functions. (**B**). Western blot analysis demonstrating the absence of ATG5-ATG12 conjugation in *Arabidopsis* transgenic line expressing ATG5^K128R^ -TAP. α-actin was used for protein loading control. (**C**). The ATG5^K128R^ mutant fails to restore autophagy in *ATG5*-deficient cells. Confocal microscopy demonstrating accumulation of autophagic bodies (white arrowheads) in the vacuole of *Arabidopsis* protoplasts. Protoplasts were isolated from *atg5-1*
*Arabidopsis* plants expressing GFP-ATG8a and transformed with plasmids encoding ATG5-TAP, ATG5^K128R^-TAP, ATG5-TagRFP, and ATG5^K128R^-TagRFP. To induce autophagy, protoplasts were treated with 5µM AZD and 1 µM ConA for 24 h prior to imaging. Scale bars, 10 µm. (**D**). Representative pictures of two-month-old *Arabidopsis* plants under normal growth conditions (16 h 150 μM light, 22 °C). The ATG5-TAP fusion protein is expressed under the native ATG5 promoter complements the autophagy-deficient phenotype of the *atg5-1* mutant, unlike the ATG5^K128R^-TAPa mutant.

**Figure 2 ijms-24-12300-f002:**
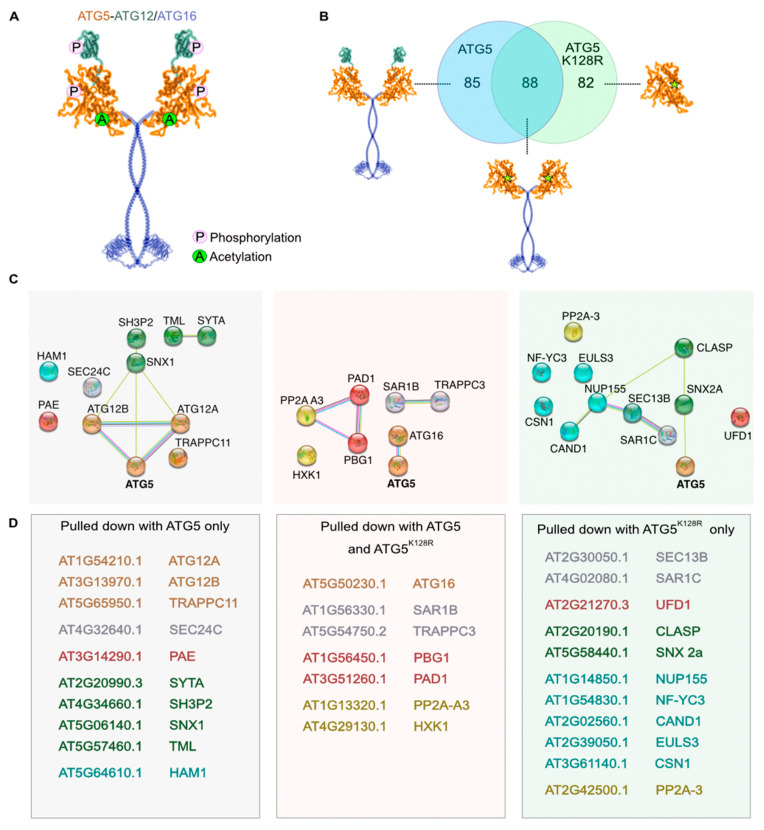
ATG5 and ATG5^K128R^ interactomes overlap only partially. (**A**). Post-translational modifications identified using LC-MS/MS. Phosphorylation of Lys183 was detected for both ATG5 and ATG5^K128R^, while phosphorylation of Ser187 was detected for ATG5 only. Additionally, we observed phosphorylation of Ser7 for ATG12A. (**B**). Venn diagram displaying partially overlapping interactome networks of ATG5 and ATG5^K128R^. Eighty-five proteins were found in ATG5 pull-down samples only, indicating that those might be interactors of either complete complex or of ATG12. Eighty-two proteins were pulled down only with ATG5^K128R^ indicating that they might be interacting with the individual ATG5 form, potentially the region masked in the ATG5-ATG12 conjugate. Eighty-eight proteins were shared between ATG5 and ATG5^K128R^ pull-downs, suggesting that those might be interactors of the minimal complex. (**C**). STRING analysis of the selected interactors pulled down with ATG5 only, ATG5^K128R^ only and with both types of baits. (**D**). List of selected proteins pulled down with ATG5 only, ATG5^K128R^ only and with both types of baits.

**Figure 3 ijms-24-12300-f003:**
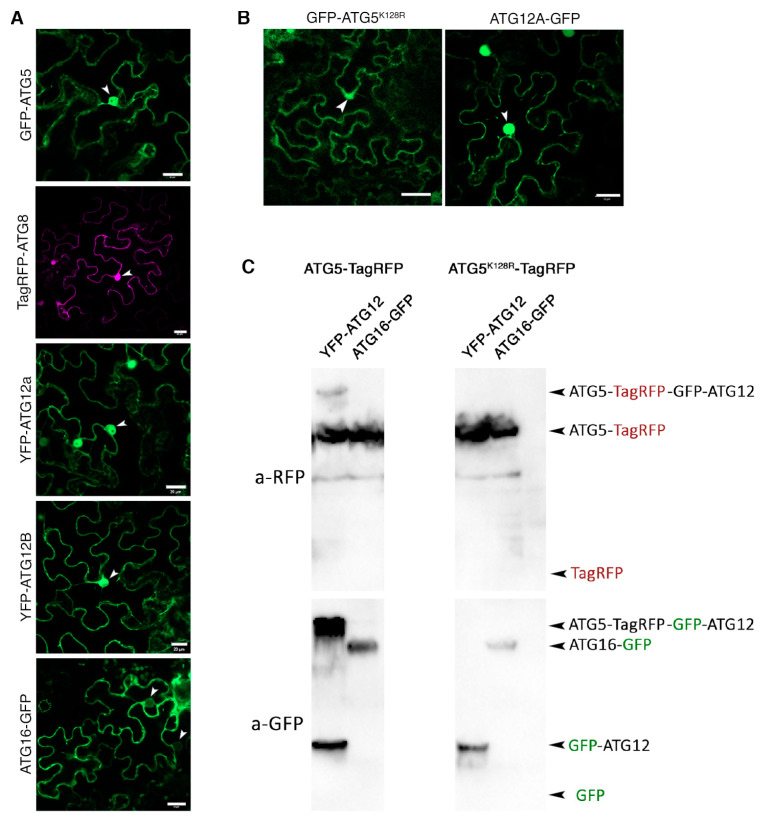
Conjugation is dispensable for nuclear localization of *Arabidopsis* ATG5 and ATG12. (**A**). Localization of fluorescently labelled ATG5, ATG8, ATG12A, ATG12B and ATG16 transiently expressed in *Nicotiana benthamiana* epidermal leaf cells. White arrowheads indicate nuclei position. Scale bar, 20 mm. (**B**). Localization of fluorescently labelled conjugation-incompetent protein forms of ATG5 and ATG12 transiently expressed in *N. benthamiana* epidermal leaf cells. Both, ATG5^K128R^ and ATG12-GFP are detectable in the nuclei, indicating that conjugation is dispensable for nuclear localization of these proteins. White arrowheads indicate nuclei position. Scale bar, 20 mm. (**C**). Western blot analysis of total protein extracts from *N. benthamiana* leaves expressing fluorescently labelled ATG5, ATG12 and ATG16 confirms formation of the covalent ATG5-ATG12 conjugate if the WT, but not K128R mutant of ATG5 is expressed, and shows no presence of the free fluorescent tag in the samples.

## Data Availability

The raw LC-MS/MS data is available on the Zenodo: 10.5281/zenodo.8199375.

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
