# Peer review of "Interactome of Arabidopsis ATG5 Suggests Functions beyond Autophagy"

_ijms, 2023, doi:10.3390/ijms241512300_

Round 1

Reviewer 1 Report

Elander et al conducted a study to identify the interactome of wild-type ATG5 and its autophagy-inactive substitution mutant, ATG5K128R.  The manuscript provides information and can be accepted in its present form. I would only suggest authors to add the novelty of the work, if there is any to the abstract and introduction. Also, please add the future prospect of the study to the abstract. Please add detailed methodology in 4.5.3 so that it can followed by future researchers.  

Author Response

We would like to thank all reviewers for the constructive comments and suggestions. Please find our answers below.

Comment 1. I would only suggest authors to add the novelty of the work, if there is any to the abstract and introduction.

Answer 1:  Thank you for the suggestion, we introduced edits into the second paragraph of abstract and the last paragraph of the introduction.

Comment 2. Also, please add the future prospect of the study to the abstract.

Answer 2:  Thank you for the suggestion, we modified the last sentence of the abstract.

Comment 3. Please add detailed methodology in 4.5.3 so that it can followed by future researchers.

Answer 3:  Thank you for the comment, we introduced more detailed description of the procedure for verification of transgenic lines.

Reviewer 2 Report

Dear Authors,

In this manuscript, the authors investigated the role of autophagy-related genes, ATG5 and ATG7, in Arabidopsis thaliana and their impact on agriculturally important traits. The researchers previously demonstrated that upregulation of these genes positively influenced traits such as biomass, yield, longevity, and stress tolerance.  Now, the research described in this MS provides valuable insights into the multifaceted roles of autophagy-related genes in plants showing that plant autophagy-related proteins play roles that go beyond autophagy.

The topic is of interest, and well written, and I have only small comments.

1. The way how plants were grown (4.2 section) should be better described, including temperature, relative humidity and light intensity. 

2. “N. benthamiana plants were grown in 8 cm3 filled with soil S-Jord”. Are you referring to pots? If yes please make it clearer. Same is valid for 4.5.3 section

English is fine. Only minor editing of English language is required

Author Response

We would like to thank all reviewers for the constructive comments and suggestions. Please find our answers below.

Comment 1. The way how plants were grown (4.2 section) should be better described, including temperature, relative humidity and light intensity.

Answer 1: thank you for the comment. We added the information about plant growth conditions to the chapter.

Comment 2. “N. benthamiana plants were grown in 8 cm3 filled with soil S-Jord”. Are you referring to pots? If yes please make it clearer.

Answer 2: thank you for the comment. We corrected the sentence.

Comment 3. Same is valid for 4.5.3 section

Answer 3: thank you for the comment. We corrected the sentence.

Comment 4. English is fine. Only minor editing of English language is required

Answer 4: We took into consideration this comment and the new version of the manuscript was edited according to recommendations of a native English speaker.

Reviewer 3 Report

The manuscript titled “Interactome of Arabidopsis ATG5 Suggests Functions beyond Autophagy”. The authors have used the affinity purification followed by LC-MS/MS to identify the interactome of wild type ATG5 and its autophagy -inactive substitution mutant, ATG5K128R. Proteins identified using WT ATG5 for autophagy, and mutant ATG5K128R for minimal complex (ATG5-ATG16) formation. In addition, authors have demonstrated the partial overlap between autophagy-competent and autophagy-incompetent ATG5K128R interactomes reveals molecular ATG5 partners.

However, the manuscript still needs improvisation. Therefore, I recommend the authors to incorporate the following points in the manuscript for further consideration.

Check the figure 1 and figure 2 descriptions/caption carefully.

Figures should be inserted to the main text close to the first citation and must be numbered following their number of appearance (Figure 1, Figure 2 etc.).

In Figure 2, do not start the lines with numerals. Eg. 85 proteins should be eighty five proteins…

Authors are advised to provide an overall framework figure listing all the analysis and steps done in this paper in materials and methods.

Plant scientific names should be in full form in first mention rest should be abbreviated. Authors should revise this throughout the manuscript. Eg. In figure 3 caption Nicotiana benthamiana and next mention should be N. benthamiana

STRING analysis (4.4.4) should be explained more. Why have authors chosen medium confidence instead of high (0.700) or highest (0.900) confidence score?

The materials and methods section looks shallow. Improve it with more detailed analysis.

The discussion section needs to be revised clearly. especially 3.1 section last paragraph.

Authors should provide a few lines about future perspectives or hypotheses about the study. It will be useful to the readers for ease of understanding.

Author Response

We would like to thank all reviewers for the constructive comments and suggestions. Please find our answers below.

Comment 1: Check the figure 1 and figure 2 descriptions/caption carefully.

Answer 1: thank you, we have revised the Figure legends. Please see the revised version of the manuscript

Comment 2. Figures should be inserted to the main text close to the first citation and must be numbered following their number of appearance (Figure 1, Figure 2 etc.).

Answer 2: as much as we agree with the reviewer on this point, the figures were inserted into the text after manuscript formatting at MDPI. We corrected the figures placement in the revised formatted text.

Comment 3. In Figure 2, do not start the lines with numerals. Eg. 85 proteins should be eighty five proteins…

Answer 3: thank you for bringing it up, we introduced corresponding corrections.

Comment 4. Authors are advised to provide an overall framework figure listing all the analysis and steps done in this paper in materials and methods.

Answer 4: thank you for the suggestion, we considered including such figure during the manuscript preparation. However, we eventually decided against including it, since the implemented protocols are commonly used.

Comment 5. Plant scientific names should be in full form in first mention rest should be abbreviated. Authors should revise this throughout the manuscript. Eg. In figure 3 caption Nicotiana benthamiana and next mention should be N. benthamiana

Answer 5: thank you! We corrected the text accordingly, keeping complete Latin name mentioned only once/manuscript section.

Comment 6. STRING analysis (4.4.4) should be explained more. Why have authors chosen medium confidence instead of high (0.700) or highest (0.900) confidence score?

Answer 6: thank you for this suggestion, we edited the corresponding materials and methods section.

Comment 7. The materials and methods section looks shallow. Improve it with more detailed analysis.

Answer 7: we added detailed information based on the comments from all three reviewers and expanded the information provided in the Table S4.

Comment 8. The discussion section needs to be revised clearly. especially 3.1 section last paragraph.

Answer 8: we edited the discussion chapter and rephrased the last paragraph.

Comment 9. Authors should provide a few lines about future perspectives or hypotheses about the study. It will be useful to the readers for ease of understanding.

Answer 9: thank you for this suggestion. We edited the abstract, the last paragraph of the introduction and the conclusions chapter.